chemical biology/synthetic chemistry

cyclic peptides, copper catalysis, peptide macrocyclization, peptide cyclase 1 (PCY1), asymmetric ligand design, metallopeptides

**Author for correspondence:**
Amanda G. Jarvis
e-mail: amanda.jarvis@ed.ac.uk

This article has been edited by the Royal Society of Chemistry, including the commissioning, peer review process and editorial aspects up to the point of acceptance.

# Macrocylases as synthetic tools for ligand synthesis: enzymatic synthesis of cyclic peptides containing metal-binding amino acids

Richard C. Brewster, Irati Colmenero Labeaga, Catriona E. Soden and Amanda G. Jarvis

EaStCHEM School of Chemistry, University of Edinburgh, Joseph Black Building, David Brewster Rd, Edinburgh EH9 3FJ, Scotland

RCB, 0000-0002-3932-9570; AGJ, 0000-0002-6414-0497

Improving the sustainability of synthesis is a major goal in green chemistry, which has been greatly aided by the development of asymmetric transition metal catalysis. Recent advances in asymmetric catalysis show that the ability to control the coordination sphere of substrates can lead to improvements in enantioselectivity and activity, in a manner resembling the operation of enzymes. Peptides can be used to mimic enzyme structures and their secondary interactions and they are easily accessible through solid-phase peptide synthesis. Despite this, cyclic peptides remain underexplored as chiral ligands for catalysis due to synthetic complications upon macrocyclization. Here, we show that the solid-phase synthesis of peptides containing metal-binding amino acids, bipyridylalanine (**1**), phenyl pyridylalanine (**2**) and *N,N*-dimethylhistidine (**3**) can be combined with peptide macrocylization using peptide cyclase 1 (PCY1) to yield cyclic peptides under mild conditions. High conversions of the linear peptides were observed (approx. 90%) and the Cu-bound cyclo(FSAS(**1**)SSKP) was shown to be a competent catalyst in the Friedel-Crafts/conjugate addition of indole. This study shows that PCY1 can tolerate peptides containing amino acids with classic inorganic and organometallic ligands as side chains, opening the door to the streamlined and efficient development of cyclic peptides as metal ligands.

## 1. Introduction

The use of catalysts in place of stoichiometric reagents is one of the 12 Principles of Green Chemistry [1]. Asymmetric catalysis allows

the selective synthesis of enantiopure products from racemic starting materials and thus minimizes waste via reducing the production of unwanted stereoisomers. Enzymes can be considered the ultimate asymmetric catalysts, as they are highly effective and selective catalysts that work under mild conditions in benign solvents. Chemists have been inspired by nature's designs, creating biomimetic catalysts which try to reproduce the active site of an enzyme. However, simple metal catalysts lack the complex structural environment and secondary interactions encountered in an enzyme, which are often responsible for their extraordinary function. Advances in catalysis have highlighted the benefits of introducing these features to chiral ligands for improving selectivity in asymmetric catalysis [2,3]. For example, the use of hydrogen bonding motifs to control substrate bindings [4], or non-covalent–π interactions to control selectivity [5]. One drawback of these developments is the economic and environmental cost of increasing the complexity of the synthesis of the catalyst. Alternatively, artificial metalloenzymes, proteins functionalized with an achiral catalyst have been demonstrated to be effective tools for numerous asymmetric metal-catalysed reactions not seen in nature [6]. However, they have not yet been applied outside academic laboratories as they are expensive, and though genetic optimization allows access to large libraries this is time consuming and requires biological expertise.

In between these two extremes lie metallopeptides. Peptides are able to form a number of conformationally constrained structures, α-helices [7,8], β-loops and sheets [9–13], coiled coils [14], and macrocycles [15–18], which have been shown to be to be beneficial in asymmetric catalysis [19–22]. As well as finding use in catalysis, metallopeptides have been studied extensively in medicinal chemistry. For example, metal binding to antimicrobial peptides has been shown to influence their activity [23], and a number of metallopeptides have shown interesting antifungal and anti-cancer activity [24,25]. Additionally, metallopeptides have been designed for use as imaging agents [26], metal sensors [27,28] and metal chelation [29].

One of the advantages of studying metallopeptides is that well-established routes to large libraries of peptide variants, including unnatural amino acids, are available using solid-phase peptide synthesis (SPPS). These methods have been successfully automated providing high-throughput synthesis of peptides for screening purposes [30,31]. However, key challenges still arise in the synthesis peptides, including efficient access to enantiomerically pure amino acids and cyclization. Cyclic peptides are more rigid and confined than their linear counterparts; however, they have been relatively underexplored as metallopeptide scaffolds [15]. This is primarily due to synthetic difficulties in their preparation as the cyclization step has several limitations [32,33]. These include epimerization at the site of cyclization [34], competing oligomerization [35] and the requirement of protecting groups to achieve selectivity adding subsequent synthetic steps [36,37]. Nature uses enzymes to catalyse the cyclization of peptides and control selectivity [38,39]. The broad substrate scope of enzyme macrocyclases such as PatGmac [40–42] from the patellamide biosynthetic pathway and peptide cyclase 1 (PCY1) [43] have made them attractive tools in the synthesis of cyclic peptides. PCY1 is a macrocyclase involved in the biosynthesis of segetalins and has been shown to be the fastest macrocyclase known to date ($k_{cat}/K_m = 830\,000\ \text{M}^{-1}\,\text{s}^{-1}$) as well as a highly promiscuous enzyme [43,44]. Recent work by Ludewig *et al.* demonstrated that PCY1 could cyclize simpler synthetically accessible presegetalins (linear precursors of segetalins) with a shortened 3-residue C-terminal recognition sequence at an efficient rate, and a wide range of peptides containing both canonical and non-canonical amino acids were tolerated as substrates [43].

To date, the development of macrocyclases has focused on therapeutic purposes such as novel antibiotics, and thus the substrate scope has focused on the presence of medicinally relevant side chains. Expanding the scope to include metal-binding side chains would be beneficial not only for medical applications, i.e. metallodrugs and imaging agents, but also for applications including catalysis and metal remediation. Herein, we present our work to expand the substrate scope of PCY1 to include peptides containing metal-binding side chains, which combined with SPPS would open the door to the efficient synthesis of cyclic peptides containing metal-binding amino acids.

# 2. Material and methods

All synthetic procedures were carried out using standard synthetic procedures and are detailed in full in the electronic supplementary material alongside supporting characterization. Linear peptides were synthesized using standard SPPS conditions with Fmoc-protected amino acids. The macrocyclase, PCY1, was expressed and purified as described by Ludewig *et al.* [43].

**Figure 1.** Unnatural amino acids used in the synthesis of cyclic peptides.

Peptide cyclization reactions were carried out in buffer (20 mM Tris, 100 mM NaCl, 5 mM DTT, pH 8.5). A 5 mM stock of peptide in $H_2O$ (where $\varepsilon = 17\,940\,M^{-1}\,cm^{-1}$ or $10\,273\,M^{-1}\,cm^{-1}$, for FSAS(**1**)SSKPFQA and FSAS(**2**)SSKPFQA, respectively, at 280 nm) and a 0.10 mM stock of PCY1 in buffer were prepared. The appropriate amounts of PCY1 solution, peptide solution and buffer were added to a Falcon tube, to give a final concentration of 2 μM PCY1 and 500 μM peptide. The reaction mixture was incubated at 30°C, 200 r.p.m. overnight. The reaction was quenched in TFA (1.7%), centrifuged for 5 min to remove precipitated protein and filtered (0.45 μm). The peptide solutions were analysed by LC-MS.

Copper-binding experiments were performed on a 100 μl scale in a BRAND® micro UV-cuvette. A 20 μM solution of either bipyridine or cyclo(FSAS(**1**)SSKP) in 20 mM 2-(N-morpholino)ethanesulfonic acid (MES), 50 mM NaCl, pH 6.00 was titrated with aliquots of a $Cu(NO_3)_2$ solution (500 μM in water), and the UV spectra recorded between 240 and 350 nm. The Friedel-Crafts/conjugate addition of indole was performed as previously described in the literature [45].

# 3. Results and discussion

## 3.1. Amino acid synthesis

In order to explore the types of metal-binding side chains that could be tolerated by PCY1, three different amino acids were chosen: 2,2-bipyridylalanine, **1a**, 2-phenylpyridylalanine, **2a** and N,N-dimethylhistidine, **3a** (figure 1), representing classic inorganic and organometallic ligands, while avoiding air-sensitive ligands such as phosphines which would complicate peptide synthesis. The modified amino acids were synthesized with the fluorenylmethyloxycarbonyl (Fmoc) protecting group (**1–3b**), compatible with Fmoc-SPPS strategies. Amino acids **1b** and **2b** were synthesized via chiral alkylation of N-(diphenylmethylene)glycine t-Butyl ester with the respective aryl bromides, in the presence of the Lygo phase transfer catalyst, N-(9-anthracenylmethyl)cinchonindinium chloride [46]. The desired amino acids were obtained in good yield and enantioselectivity (approx. 85% ee). Interestingly variation of the base, solvent and temperature (0 to −78°C) did not improve the enantioselectivity but did cause a decrease in yield. This method represents an improvement on the previously reported chiral synthesis of **1b** which used re-crystallization in the presence of chiral acids to obtain the enantiopure amino acid from a racemic mixture [47,48].

Amino acid **3b** was synthesized via methylation of the commercially available N-Boc L-histidine **4** followed by protecting group exchange (scheme 1). Unexpectedly, Fmoc protection of compound **3a** proceeded to give a mixture of the expected product, **3b**, and a product containing two Fmoc groups (1.1 eq. Fmoc-Cl and 10% $Na_2CO_{3(aq)}$). Using a weaker base ($NaHCO_3$) or increasing the equivalents of Fmoc-Cl (1.1–4.4 eq.) still gave a mixture of products [49].[1] Purification by reverse phase chromatography gave a small amount of both the single and bis-Fmoc containing product. NMR analysis shows that the two Fmoc groups are equivalent suggesting that in the bis-Fmoc product both Fmoc residues are bound to the nitrogen of the amino acid giving **3c**, as opposed to the formation of the Fmoc anhydride on the carboxylic acid. IR spectroscopy supports the identity of **3c** as no C=O

---

[1]Since the submission of this manuscript, Albrecht, Paradisi and co-workers also reported difficulties with the direct synthesis of the Fmoc-protected imidazolium amino acid and showed that through sequential protecting group exchange compound **3b** can be synthesized in good yields.

**Scheme 1.** Synthesis of *N,N*-dimethylhistidine for SPPS.

stretch between 1750 and 1825 cm$^{-1}$ is observed, which would be expected if the compound was the anhydride of the acid. This unusual result suggests that the imidazolium activates the Fmoc-Cl to nucleophilic attack of the carbamate. Compound **3c** was taken forward into SPPS to see if deprotection of the double protected amino acid would proceed smoothly under standard conditions to give the desired peptide.

## 3.2. Peptide synthesis

With the Fmoc-protected amino acids in hand, SPPS could be used to provide the linear precursor peptides for cyclization. The presegetalin FSASYSSKP-FQA, containing FQA as the C-terminal recognition sequence, was previously shown to be a substrate for PCY1, resulting in segetalin F on cyclization [43,50]. The tyrosine residue was identified as an ideal location to introduce different amino acids as it is already sterically large and far away from the cyclization site. Using a standard Fmoc-SPPS protocol, peptides FSAS(**1**)SSKP-FQA, FSAS(**2**)SSKP-FQA and FSAS(**3**)SSKP-FQA alongside FSASYSSKP-FQA were successfully synthesized. LC-MS showed that the crude peptides FSAS(**1**)SSKP-FQA, FSAS(**2**)SSKP-FQA and FSASYSSKP-FQA were obtained in excellent purity (greater than 90%) and were used without further purification.

The macrocylase PCY1 was expressed in *E. coli* BL21 and purified by Ni-affinity chromatography according to the literature [43]. SDS-PAGE gel of the purified protein showed the presence of two main bands: the desired enzyme band, at 83 kDa, and another band at 27 kDa (electronic supplementary material, figure S8). The identity of this secondary band was not determined, and the purity was considered sufficient to proceed with the cyclization. A brief optimization of the cyclization conditions with both FSASYSSKP-FQA and FSAS(**1**)SSKP-FQA was conducted (see electronic supplementary material, table S1 and S2). A cyclase concentration of 2 µM and a peptide concentration of 500 µM were found to give the optimal performance providing conversions of over 90% to the cyclic peptide while maintaining low-catalyst loadings (see electronic supplementary material for further details). The remaining material was predominantly the truncated peptide, FSASYSSKP or FSAS(**1**)SSKP, arising from the cleavage of the recognition sequence, FQA, from the linear peptide (figure 2). The presence of the bulkier bipyridylalanine did not affect the high conversion of the reaction (94% cyclo(FSAS(**1**)SSKP) versus 90% cyclo(FSASYSSKP)), highlighting that the PCY1 can accept bulkier residues in that position. Using the optimized conditions on a 13 µmol scale (17 mg), cyclo(FSAS(**1**)SSKP) and cyclo(FSAS(**2**)SSKP) were obtained after purification with isolated yields of 35 and 27% respectively, which compared favourably to the obtained yield of 18% for cyclo(FSASYSSKP). The low yield compared to the conversion is attributed to the difficulty in isolating the peptide from dilute buffer solutions. The identity of the peptides was supported by NMR characterization (electronic supplementary material, figure S12). In the 1D $^1$H NMR spectra of cyclo(FSASYSSKP), two stable conformers were observed on the NMR timescale in approximately 7 : 3

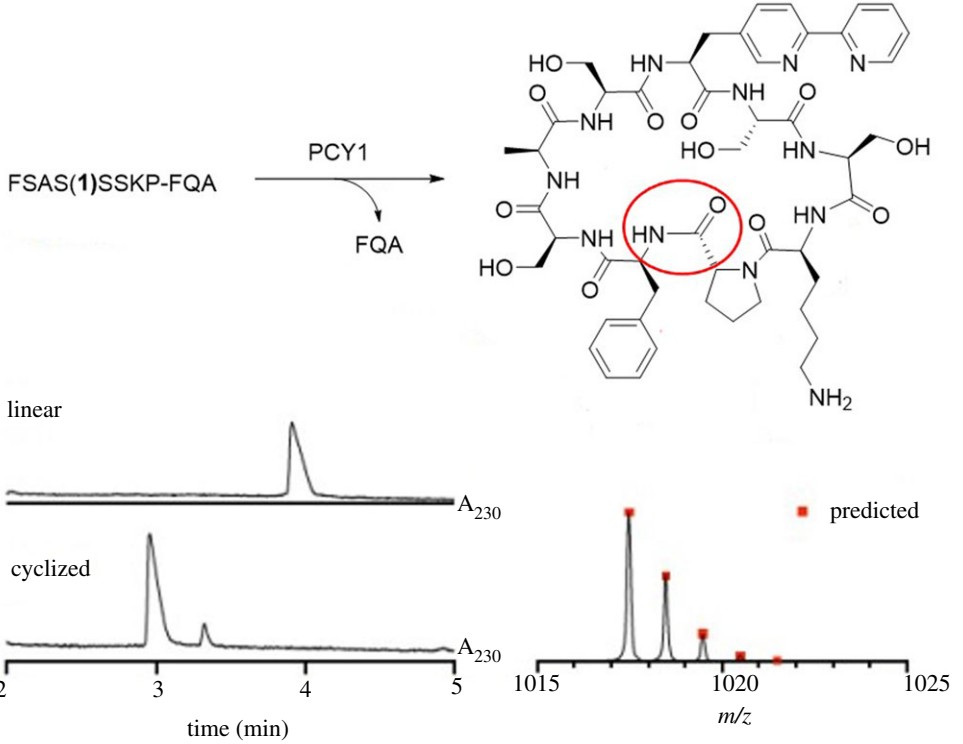

**Figure 2.** Top: reaction scheme for the cyclization of FSAS(**1**)SSKP-FQA with PCY1 with the new bond highlighted in red. Conditions: PCY1 (2 μM) and the linear peptide (500 μM) in buffer (20 mM Tris, 100 mM NaCl, 5 mM DTT, pH 8.5) were incubated overnight at 30°C, 200 r.p.m. Bottom left: UPLC chromatograms for the linear peptide (top) and crude cyclised product (bottom, where the cyclo(FSAS(**1**)SSKP) is at $R_T = 3.0$ min, and the truncated linear peptide FSAS(**1**)SSKP at $R_T = 3.4$ min). Bottom right: mass spectrum for the cyclized peptide showing excellent alignment with the predicted mass and isotopic distribution for the [M + H]$^+$ ion.

**Scheme 2.** Scheme showing the products in the synthesis of FSAS(**3**)SSKP-FQA.

ratio. However, both cyclo(FSAS(**1**)SSKP) and cyclo(FSAS(**2**)SSKP) gave less conclusive spectra, with a complex aromatic region suggesting multiple conformers/rotamers around the bipyridine and phenylpyridyl side chains.

The linear peptide FSAS(**3**)SSKP-FQA was prepared on a 4 μmol (approx. 4 mg) scale using **3c** as the protected amino acid. MALDI FT ICR-MS analysis confirmed formation of the linear peptide; however, it also showed the presence of a truncated product (**7**) with a mass corresponding to the addition of a CO and loss of 2H from the intermediate peptide **6** (see electronic supplementary material, figure S9 and S10; scheme 2). Despite the presence of this side product, macrocyclization was attempted without further purification. Treatment of the FSAS(**3**)SSKP-FQA with PCY1 gave cyclo(FSAS(**3**)SSKP) as confirmed by mass spectrometry; however, due to the small scale of the reaction isolation of the cyclic peptide was not attempted. The presence of a species with a mass corresponding to hydrolysis of the side product was also identified by mass spectrometry after the enzymatic step, and none of the initial side product remained (electronic supplementary material, figure S11).

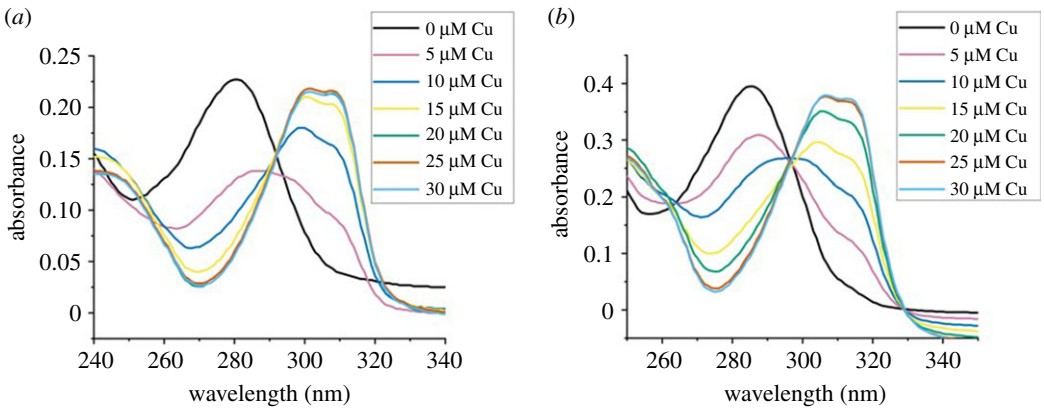

**Figure 3.** Titration of (*a*) bipyridine (bipy) and (*b*) cyclo(FSAS(**1**)SSKP) (20 µM in 20 mM 2-(N-morpholino)ethanesulfonic acid (MES), 50 mM NaCl, pH 6.0) with increasing concentrations of Cu(NO₃)₂: 0 µM (black), 5 µM (pink), 10 µM (dark blue), 15 µM (yellow), 20 µM (green), 25 µM (orange) and 30 µM (light blue).

Overall, it was shown that PCY1 would accept peptides containing the metal-binding amino acids, **1–3**, while retaining the level of conversion observed with the native substrate. This provides a promising synthetic method for the synthesis of cyclic metallopeptides.

## 3.3. Metal binding and catalysis

Metallopeptides are attractive compounds for a variety of applications from diagnostic probes and metallodrugs to chiral catalysts. Having shown that PCY1 could accept amino acids containing potential metal-binding sites in their side chains, we were interested in testing if the resulting cyclic peptides would act as metal ligands. The coordination chemistry of bipyridine has been extensively studied using UV-visible spectroscopy which is ideal to use with peptide ligands [51]. The ability of FSAS(**1**)SSKP to bind copper(II) was shown by comparison to bipyridine (bipy) using UV-visible studies (figure 3*a* versus *b*). On addition of Cu(NO₃)₂ to a solution of cyclo(FSAS(**1**)SSKP) in MES buffer (20 µM), a characteristic red shift in the $\pi-\pi^*$ transition of the bipyridyl moiety from $\lambda_{\max} = 280$ nm to 301 nm was observed indicating Cu(II) binding (figure 3*b*). Saturation of the bipy-binding site was achieved after the addition of 1 equivalent of Cu(NO₃)₂ implying a 1:1 metal to ligand ratio. The similarity between both UV-vis spectra suggests that the peptide is not influencing the metal coordination chemistry or electronic properties of the bipyridine. The addition of Cu(NO₃)₂ to cyclo(FSAS(**1**)SSKP) was followed by mass spectrometry and showed the formation of a new species with a mass of 1078.39 Da which corresponds to a 1:1 copper complex of cyclo(FSAS(**1**)SSKP) (electronic supplementary material, figure S13).

The Friedel-Crafts/conjugate addition of indole shown in scheme 3 was chosen as a benchmark reaction to test for catalytic activity and asymmetric induction using cyclo(FSAS(**1**)SSKP) as a ligand [45]. The Cu(II) complex of cyclo(FSAS(**1**)SSKP) was shown to be a competent catalyst for the reaction, giving similar yields to those obtained with bipyridine as a ligand (53% versus 58% for Cu(bipy)(NO₃)₂, see electronic supplementary material, table S3). Disappointingly, essentially no asymmetric induction was observed in this reaction in the presence of the peptide (e.r. 52:48). This is perhaps unsurprising, as the choice of peptide structure was driven by the substrate scope of the macrocyclase PCY1 and not optimized for asymmetric induction. The lack of asymmetric induction suggests that the copper-bipyridine region of the peptide is exposed to the solvent, as opposed to within a more defined peptide environment. This is supported by the one-dimensional ¹H NMR characterization of cyclo(FSAS(**1**)SSKP) which showed a complex pattern in the aromatic region of the NMR spectrum suggesting a number of stable conformations with subtly different environments of the bipyridine are present, which presumably contribute to the lack of stereo-differentiation observed in the catalysis. In order to fully understand the structure of the cyclic peptides and to aid future ligand optimization a full structural study by NMR is currently underway.

Many examples of successful metal-binding peptides include multiple ligands to reduce flexibility around the metal centre by creating multicyclic structures. Therefore, to improve the enantioselectivity of cyclo(FSAS(**1**)SSKP), an additional bipyridyl side chain or other metal-binding amino acids such as histidine could be introduced to rigidify the structure while also leaving vacant coordination sites for

**Scheme 3.** Copper catalysed Friedel-Crafts/conjugate addition of indole.

catalysis. We expect future studies exploring a more diverse range of peptide sequences including multiple metal-binding amino acids, to lead to more promising asymmetric catalysts.

# 4. Conclusion

In this study, we have shown that the metal-binding amino acids **1** to **3** were compatible with SPPS and that the substrate scope of PCY1 could be expanded to include peptides containing unnatural amino acids with metal ligands as side chains, including both aromatic and charged groups. This resulted in the synthesis of three cyclic peptides in excellent conversions: cyclo(FSAS(**1**)SSKP), cyclo(FSAS(**2**)SSKP) and cyclo(FSAS(**3**)SSKP), under mild conditions and avoiding the problems traditionally associated with cyclic peptide synthesis. The metal-binding properties of cyclo(FSAS(**1**)SSKP) with copper(II) were studied by UV-vis spectroscopy and MS, and showed stoichiometric binding in a 1:1 ratio. The copper complex of cyclo(FSAS(**1**)SSKP) was shown to be a competent catalyst in the Friedel-Crafts/conjugate addition of indole albeit without providing asymmetric induction. Overall, this work provides proof of concept that combining enzymatic macrocyclization with SPPS provides a streamlined methodology that would be suitable for preparing libraries of cyclic metallopeptides for screening in both asymmetric catalysis and medicinal chemistry. Future work will focus on obtaining the structures of the cyclic peptides to allow optimization of the sequence for improved asymmetric catalysis.

Data accessibility. Experimental data supporting this article have been uploaded as part of the electronic supplementary material. Additional data files are available on the University of Edinburgh DataShare database at http://doi.org/10.7488/ds/3157.

The data are provided in the electronic supplementary material [52].

Authors' contributions. A.G.J. and R.B. conceived and designed the study. R.B., I.C. and C.E.S. carried out the experimental work, data collection and analysis. A.G.J. wrote the manuscript with contributions from R.B., I.C. and C.E.S. All authors have agreed to the final version.

Competing interests. We declare we have no competing interests.

Funding. A.G.J. and R.C.B. were funded through a UKRI Future Leaders Fellowship (grant no. MR/S017402/1) awarded to A.G.J. I.C.L. and C.E.S. conducted this work as part of their MChem studies at the University of Edinburgh.

Acknowledgements. We would like to thank Professor Jim Naismith for the gift of the PCY1 plasmid and Dr Hannes Ludewig for advice on producing PCY1. We thank Professor Alison Hulme for access to prep-HPLC/UPLC instruments, Dr Faye Cruickshank and the Mass Spectrometry Service (SIRCAMS) in the School of Chemistry, University of Edinburgh for MALDI and MS analysis, and Dr Juraj Bella for help with the NMR spectra of the peptides.

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
