## [Peer Review File · Royal Society Open Science]

Review History

RSOS-211098.R0 (Original submission)

Review form: Reviewer 1

Is the manuscript scientifically sound in its present form?

Yes

Are the interpretations and conclusions justified by the results?

No

Is the language acceptable?

Yes

Do you have any ethical concerns with this paper?

No

Have you any concerns about statistical analyses in this paper?

No

Recommendation?

Major revision is needed (please make suggestions in comments)

Comments to the Author(s)

The manuscript by Jarvis and co-workers describes the chemo-enzymatic synthesis of cyclic peptides containing non-natural amino acids able to bind transition metals. The corresponding non-natural amino acids were also synthesized with the suitable protection to be included in SPPS, which was combined with a biocatalytic process for the preparation of the macrocyclic peptides. The binding of Cu(II) was monitored by UV-vis spectroscopy titration. The Cu(II) complex is a catalyst for Friedel-Crafts alkylation though lacking asymmetric induction. The work is well performed and the manuscript is clear (albeit some typos and grammar errors that should be corrected in the revised version). The lack of asymmetric induction suggests that the design of the peptidic ligand is not optimal. Actually, the similarity between the UV titration of the peptide and simply bipy already suggests that the coordination geometry is very similar. I would support acceptance of a revised version of the manuscript following some indications.

1) As said before, please revise the text for typos and grammar errors.

2) The introduction is too focused in catalysis but the results on this area are extremely modest. This gives a general impression of a failure that does not reflect the merit of the overall work. The synthetic part and the metal binding are well performed and the authors must better underscore those issues in the introduction. This will give the reader a more positive impression of the overall work.

3) The synthesis of the peptides is fine, as well as those of the corresponding amino acids. However, the authors should also carry out NMR spectra of the cyclic peptides in order to get an idea about the rigidity and conformational freedom of the ligand. If too flexible, maybe the peptide is not affecting the structural space of the coordination sphere, and this is the main reason to get no stereodifferentiation.

4) The structure proposed for compound 7 must be better supported by NMR and maybe molecular modeling since this seems a highly strained bicyclic structure.

5) In the absence of stereoselective catalysis, maybe the authors could complement the metal binding behavior in a better way, by studying the complexation of other metals with all the cyclic peptides. Maybe NMR (when possible) and ESI-MS should be a good combination to demonstrate the metal binding abilities of the cyclic peptides. Peptides binding transition metals could have very interesting applications apart from catalysis (imaging, sensing, bioinorganic chemistry, detoxification, etc.).

6) From the UV-vis titrations, the authors could have obtained binding constants to better characterize the interaction. Again, since the catalysis is not really successful, the more professional characterization of the metal complexes would improve the overall draft.

7) Table 1 is somehow disappointing since Cu(II) alone seems to work better than with bipy or the corresponding macrocyclic peptide-Cu complex. The authors should use those results to get ideas on how to improve the ligands, and they must discuss that in the text. Maybe including monoligands at two different positions of the macrocycle would produce more hindered metallabicyclic complexes with some chances to create an asymmetric environment for catalysis. In any case, asymmetric catalysis using metal complexes of synthetic small peptides is an extremely challenging goal.

Overall, the work merits publication in a good journal, especially because the combination of chemoenzymatic synthesis with metal binding is somehow original and new. However, the excessive focus on the potential application on catalysis and the modest results in the catalytic assays makes the overall reading a bit disappointing. My recommendation is to better highlight the synthetic part of the work, to complement the metal binding studies and to downgrade the focus on stereoselective catalysis.

Review form: Reviewer 2 (Nicholas Turner)

Is the manuscript scientifically sound in its present form?

Yes

Are the interpretations and conclusions justified by the results?

Yes

Is the language acceptable?

Yes

Do you have any ethical concerns with this paper?

No

Have you any concerns about statistical analyses in this paper?

No

Recommendation?

Accept as is

Comments to the Author(s)

This is an excellent paper in which the cyclase enzyme PCY1 has been shown to catalyse the cyclisation of 3 peptides in ca. 90% yield. Each of these substrates contains unnatural amino acids highlighting the use of this cyclase to access non-natural cyclic peptides which are of profound biological interest. The paper is well written and merits publication in its present form.

Decision letter (RSOS-211098.R0)

Dear Dr Jarvis:

Title: Macrocyklases as synthetic tools for ligand synthesis: enzymatic synthesis of cyclic peptides containing metal-binding amino acids

Manuscript ID: RSOS-211098

The editor assigned to your manuscript has now received comments from reviewers. We would like you to revise your paper in accordance with the referee and Subject Editor suggestions which can be found below (not including confidential reports to the Editor). Please note this decision does not guarantee eventual acceptance.

Please submit your revised paper before 09-Sep-2021. Please note that the revision deadline will expire at 00.00am on this date. If we do not hear from you within this time then it will be assumed that the paper has been withdrawn. In exceptional circumstances, extensions may be possible if agreed with the Editorial Office in advance. We do not allow multiple rounds of

revision so we urge you to make every effort to fully address all of the comments at this stage. If deemed necessary by the Editors, your manuscript will be sent back to one or more of the original reviewers for assessment. If the original reviewers are not available we may invite new reviewers.

RSC Associate Editor:
Comments to the Author:
(There are no comments.)

RSC Subject Editor:
Comments to the Author:
(There are no comments.)

Reviewers' Comments to Author:

Reviewer: 1

Comments to the Author(s)

The manuscript by Jarvis and co-workers describes the chemo-enzymatic synthesis of cyclic peptides containing non-natural amino acids able to bind transition metals. The corresponding non-natural amino acids were also synthesized with the suitable protection to be included in SPPS, which was combined with a biocatalytic process for the preparation of the macrocyclic peptides. The binding of Cu(II) was monitored by UV-vis spectroscopy titration. The Cu(II)

complex is a catalyst for Friedel-Crafts alkylation though lacking asymmetric induction. The work is well performed and the manuscript is clear (albeit some typos and grammar errors that should be corrected in the revised version). The lack of asymmetric induction suggests that the design of the peptidic ligand is not optimal. Actually, the similarity between the UV titration of the peptide and simply bipy already suggests that the coordination geometry is very similar. I would support acceptance of a revised version of the manuscript following some indications.

1) As said before, please revise the text for typos and grammar errors.

2) The introduction is too focused in catalysis but the results on this area are extremely modest. This gives a general impression of a failure that does not reflect the merit of the overall work. The synthetic part and the metal binding are well performed and the authors must better underscore those issues in the introduction. This will give the reader a more positive impression of the overall work.

3) The synthesis of the peptides is fine, as well as those of the corresponding amino acids. However, the authors should also carry out NMR spectra of the cyclic peptides in order to get an idea about the rigidity and conformational freedom of the ligand. If too flexible, maybe the peptide is not affecting the structural space of the coordination sphere, and this is the main reason to get no stereodifferentiation.

4) The structure proposed for compound 7 must be better supported by NMR and maybe molecular modeling since this seems a highly strained bicyclic structure.

5) In the absence of stereoselective catalysis, maybe the authors could complement the metal binding behavior in a better way, by studying the complexation of other metals with all the cyclic peptides. Maybe NMR (when possible) and ESI-MS should be a good combination to demonstrate the metal binding abilities of the cyclic peptides. Peptides binding transition metals could have very interesting applications apart from catalysis (imaging, sensing, bioinorganic chemistry, detoxification, etc.).

6) From the UV-vis titrations, the authors could have obtained binding constants to better characterize the interaction. Again, since the catalysis is not really successful, the more professional characterization of the metal complexes would improve the overall draft.

7) Table 1 is somehow disappointing since Cu(II) alone seems to work better than with bipy or the corresponding macrocyclic peptide-Cu complex. The authors should use those results to get ideas on how to improve the ligands, and they must discuss that in the text. Maybe including monoligands at two different positions of the macrocycle would produce more hindered metallabicyclic complexes with some chances to create an asymmetric environment for catalysis. In any case, asymmetric catalysis using metal complexes of synthetic small peptides is an extremely challenging goal.

Overall, the work merits publication in a good journal, especially because the combination of chemoenzymatic synthesis with metal binding is somehow original and new. However, the excessive focus on the potential application on catalysis and the modest results in the catalytic assays makes the overall reading a bit disappointing. My recommendation is to better highlight the synthetic part of the work, to complement the metal binding studies and to downgrade the focus on stereoselective catalysis.

Reviewer: 2

Comments to the Author(s)

This is an excellent paper in which the cyclase enzyme PCY1 has been shown to catalyse the cyclisation of 3 peptides in ca. 90% yield. Each of these substrates contains unnatural amino acids highlighting the use of this cyclase to access non-natural cyclic peptides which are of profound biological interest. The paper is well written and merits publication in its present form.

Author's Response to Decision Letter for (RSOS-211098.R0)

See Appendix A.

RSOS-211098.R1 (Revision)

Review form: Reviewer 1

Is the manuscript scientifically sound in its present form?

Yes

Are the interpretations and conclusions justified by the results?

Yes

Is the language acceptable?

Yes

Do you have any ethical concerns with this paper?

No

Have you any concerns about statistical analyses in this paper?

No

Recommendation?

Accept as is

Comments to the Author(s)

The authors have fully addressed all my initial concerns. The manuscript is now ready to be published in the present format.

Decision letter (RSOS-211098.R1)

Dear Dr Jarvis:

Title: Macrocyclases as synthetic tools for ligand synthesis: enzymatic synthesis of cyclic peptides containing metal-binding amino acids

Manuscript ID: RSOS-211098.R1

It is a pleasure to accept your manuscript in its current form for publication in Royal Society Open Science. The chemistry content of Royal Society Open Science is published in collaboration with the Royal Society of Chemistry.

Yours sincerely,
Dr Ellis Wilde
Publishing Editor, Journals

RSC Associate Editor
Comments to the Author:
(There are no comments.)

RSC Subject Editor
Comments to the Author:
(There are no comments.)

Reviewer(s)' Comments to Author:
Reviewer: 1

Comments to the Author(s)
The authors have fully addressed all my initial concerns. The manuscript is now ready to be published in the present format.

Appendix A

THE UNIVERSITY of EDINBURGH
School of Chemistry

Joseph Black Building
David Brewster Road
Edinburgh EH9 3FJ
United Kingdom

Tel +44 (0) 131 650 4715
Fax +44 (0) 131 650 6453

amanda.jarvis@ed.ac.uk
www.chem.ed.ac.uk

8th Sept 2021

Dear Professor Catlow, Ms Daly and Dr Smith,

RE: Manuscript ID: RSOS-211098

“Cyclic nonapeptide segetalin F1 derivatives containing metal binding amino acid side chains: Synthesis, metal coordination studies and catalysis” by Richard C. Brewster, Irati Colmenero, Catriona E. Soden and Amanda G. Jarvis

We would like to thank the reviewers for their time and suggestions. Please find attached the revised manuscript in line with the recommendations from the reviewers, alongside a pdf highlighting the changes in yellow and a response to the reviewer's comments. We hope you find these changes satisfactory for acceptance of this manuscript for the Royal Society Open Science themed collection: *Catalysis for a sustainable future*.

We highly appreciate your time and efforts in refereeing this paper and look forward to hearing from you.

Yours sincerely,

Dr Amanda Jarvis
UKRI Future Leaders Fellow

Head of School: Professor Colin Pulham

The University of Edinburgh is a charitable body, registered in Scotland, with registration number SC005336

Response to reviewers:

Reviewer: 1

The manuscript by Jarvis and co-workers describes the chemo-enzymatic synthesis of cyclic peptides containing non-natural amino acids able to bind transition metals. The corresponding non-natural amino acids were also synthesized with the suitable protection to be included in SPPS, which was combined with a biocatalytic process for the preparation of the macrocyclic peptides. The binding of Cu(II) was monitored by UV-vis spectroscopy titration. The Cu(II) complex is a catalyst for Friedel-Crafts alkylation though lacking asymmetric induction. The work is well performed and the manuscript is clear (albeit some typos and grammar errors that should be corrected in the revised version). The lack of asymmetric induction suggests that the design of the peptidic ligand is not optimal. Actually, the similarity between the UV titration of the peptide and simply bipy already suggests that the coordination geometry is very similar. I would support acceptance of a revised version of the manuscript following some indications.

We would like to thank reviewer 1 for their thorough reading of the manuscript, suggestions and inciteful comments. Please find below our response to their detailed comments:

1) As said before, please revise the text for typos and grammar errors.

The manuscript has been proof-read and revised for typos and grammar where errors were found.

2) The introduction is too focused in catalysis but the results on this area are extremely modest. This gives a general impression of a failure that does not reflect the merit of the overall work. The synthetic part and the metal binding are well performed and the authors must better underscore those issues in the introduction. This will give the reader a more positive impression of the overall work.

The introduction has been reworked to take some of the emphasis off catalysis though the authors note that this article has been submitted to a special issue on sustainable catalysis and the goal of the authors was catalysis. We appreciate that the metallopeptides have many other valuable applications and have highlighted this both in the introduction and conclusions. Changes made to the text have been highlighted in the attached marked up copy.

3) The synthesis of the peptides is fine, as well as those of the corresponding amino acids. However, the authors should also carry out NMR spectra of the cyclic peptides in order to get an idea about the rigidity and conformational freedom of the ligand. If too flexible, maybe the peptide is not affecting the structural space of the coordination sphere, and this is the main reason to get no stereodifferentiation.

1D NMR characterisation has been included for the peptides in the ESI (Figure S12) and a brief discussion included in the text. Currently we do not have clear evidence that the peptide is interacting with the coordination sphere of the bipyridine and we agree this is likely to be the main reason why we do not obtain stereodifferentiation.

4) The structure proposed for compound 7 must be better supported by NMR and maybe molecular modeling since this seems a highly strained bicyclic structure.

On reflection, following the reviewers comments we agree that the proposed structure is unlikely (it doubly violates Bredt's rules) and have thus removed this from the discussion, and in place just highlighted the observed results we have. As noted in the text we only obtained a small amount of the imidazolium amino acid, **3**, and therefore only a small amount of the peptide. Due to the small amounts of product, no purification of the linear peptide was attempted to separate the side product and the desired linear peptide. Therefore, further analysis by NMR was not possible. This section of the text has been reworded to make the scale clearer to the reader.

5) *In the absence of stereoselective catalysis, maybe the authors could complement the metal binding behavior in a better way, by studying the complexation of other metals with all the cyclic peptides. Maybe NMR (when possible) and ESI-MS should be a good combination to demonstrate the metal binding abilities of the cyclic peptides. Peptides binding transition metals could have very interesting applications apart from catalysis (imaging, sensing, bioinorganic chemistry, detoxification, etc.).*

Additional studies on the metal binding of the peptides by MS have been carried out and the results added to the manuscript. Due to the paramagnetic nature of Cu(II), NMR studies were not carried out as similar studies in the group have shown the absence of meaningful spectra after the addition of just 20% Cu(II). We feel the inclusion of other metals is outwith the scope of this current study.

6) *From the UV-vis titrations, the authors could have obtained binding constants to better characterize the interaction. Again, since the catalysis is not really successful, the more professional characterization of the metal complexes would improve the overall draft.*

The UV titrations shown were carried out at concentrations to observe stoichiometry of metal binding and thus at concentrations far higher than the expected K_d to ensure that all metal bound immediately on addition to the bipyridine. This behaviour is borne out by the observation of 1:1 binding. Accurate K_d should be carried out at concentrations in a similar range as the expected K_d else misleading values are found (Young and Xiao, *Biochem. J.* **2021**, 478, 1085-1116). The experimental stability constant for $\text{Cu}(\text{H}_2\text{O})_6 + \text{bipy}$ to $\text{Cu}(\text{bipy})(\text{H}_2\text{O})_4$ is $\log K_1 = 8.15$ (Irving and Mellor, *J. Chem. Soc.* **1962**, 5222-5237) giving $K_1 = 1.4 \times 10^8$ and thus $K_d = 7 \times 10^{-9}$. It has been shown that a direct forward titration using ligand in the μM range (as conducted here) would give very similar curves for any K_d value above 10^{-7} (J.S. Magyer and H. A. Goodwin, *Anal. Biochem.* **2003**, 320, 39-54). Indeed using the data we currently have a K_d for Cu bipy of around 1×10^{-5} is obtained which clearly does not match the literature values and is not expected to be accurate for the reasons mentioned above. Therefore, we have not included this analysis in the revised manuscript. From the catalytic data and experimental observations, we do not expect the K_d for binding of the peptide to be very different to just bipyridine, thus to obtain an accurate K_d value a different method would need to be used.

7) *Table 1 is somehow disappointing since Cu(II) alone seems to work better than with bipy or the corresponding macrocyclic peptide-Cu complex. The authors should use those results to get ideas on how to improve the ligands, and they must discuss that in the text. Maybe including monoligands at two different positions of the macrocycle would produce more hindered metallabicyclic complexes with some chances to create an asymmetric environment for catalysis. In any case, asymmetric catalysis using metal complexes of synthetic small peptides is an extremely challenging goal.*

The table has been replaced by a figure and the relevant control reactions mentioned in the text to keep the focus more positive. A more detailed discussion of future directions has been included. *Overall, the work merits publication in a good journal, especially because the combination of chemoenzymatic synthesis with metal binding is somehow original and new. However, the excessive focus on the potential application on catalysis and the modest results in the catalytic assays makes the overall reading a bit disappointing. My recommendation is to better highlight the synthetic part of the work, to complement the metal binding studies and to downgrade the focus on stereoselective catalysis.*

Reviewer: 2

This is an excellent paper in which the cyclase enzyme PCY1 has been shown to catalyse the cyclisation of 3 peptides in ca. 90% yield. Each of these substrates contains unnatural amino acids highlighting the use of this cyclase to access non-natural cyclic peptides which are of profound biological interest. The paper is well written and merits publication in its present form.

We thank the reviewer for their time and very positive review. As two of the authors completed this work as part of their MChem degrees, this is a wonderful review to receive and pass on.